# Mechanism of Resveratrol Dimers Isolated from Grape Inhibiting ^1^O_2_ Induced DNA Damage by UHPLC-QTOF-MS^2^ and UHPLC-QQQ-MS^2^ Analyses

**DOI:** 10.3390/biomedicines9030271

**Published:** 2021-03-08

**Authors:** Qingjun Kong, Qingzhi Zeng, Jia Yu, Hongxi Xiao, Jun Lu, Xueyan Ren

**Affiliations:** 1Xi’an Key Laboratory of Characteristic Fruit Storage and Preservation, Shaanxi Engineering Laboratory for Food Green Processing and Safety Control and Shaanxi Key Laboratory for Hazard Factors Assessment in Processing and Storage of Agricultural Products, College of Food Engineering and Nutritional Science, Shaanxi Normal University, Xi’an 710119, China; Kongqj1976@snnu.edu.cn (Q.K.); zengqz1230@163.com (Q.Z.); 2College of Food Science, Southwest University, Chongqing 400700, China; yujia61@swu.edu.cn; 3Xi’an Dairy Cow Breeding Center, Xi’an 710119, China; xhx56789@163.com; 4School of Science, and School of Public Health and Interdisciplinary Studies, Faculty of Health & Environmental Sciences, Auckland University of Technology, Auckland 1142, New Zealand

**Keywords:** resveratrol dimers, antioxidative activity, singlet oxygen quenching, DNA damage

## Abstract

Resveratrol dimers have been extensively reported on due to their antioxidative activity. Previous studies revealed that resveratrol dimer has been shown to selectively quench singlet oxygen (^1^O_2_), and could protect DNA from oxidative damage. The mechanism of resveratrol dimers protecting DNA against oxidative damage is still not clear. Therefore, in this project, the reactants and products of resveratrol dimers protecting guanine from oxidative damage were qualitatively monitored and quantitatively analyzed by UHPLC-QTOF-MS^2^ and UHPLC-QQQ-MS^2^. Results showed that when guanine and resveratrol dimers were attacked by ^1^O_2_, mostly resveratrol dimers were oxidized, which protected guanine from oxidation. Resveratrol dimers’ oxidation products were identified and quantified at *m/z* 467.1134 [M-H]^−^ and 467.1118 [M-H]^−^, respectively. The resorcinol of resveratrol dimers reacted with singlet oxygen to produce p-benzoquinone, protecting guanine from ^1^O_2_ damage. Therefore, it is hereby reported for the first time that the resorcinol ring is the characteristic structure in stilbenes inhibiting ^1^O_2_ induced-DNA damage, which provides a theoretical basis for preventing and treating DNA damage-mediated diseases.

## 1. Introduction

Oxidative stress refers to the condition in which the oxidative and anti-oxidative effects in the body are out of balance, and free radicals produce negative effects. It may cause metabolic dysregulation or oxidative damage to cells, tissues, and organs [1]. Mediators of oxidative stress, reactive oxygen species (ROS), include superoxide radical anion (O_2_^•^^−^), hydroxyl radical (·OH), hydrogen peroxide (H_2_O_2_), and ^1^O_2_ [2]. ^1^O_2_, as an excited oxygen molecule (^1^Δ_g_), can be produced by biologically relevant processes in vivo, and also by photosensitized oxidations in vitro. This can occur with two of the optogenetic sensitizers, aryloxazinones and thiopurines, which generate ^1^O_2_ at a specific wavelength of UV irradiation. ^1^O_2_ participates in a variety of physiological and pathological processes [3,4]. It is a common source of damage for many biomolecules: lipids, protein, and DNA [5,6,7].

DNA is essential in assisting cellular function, and a molecule of great biological significance. Oxidative damage of DNA has been identified as a key factor in the onset and development of numerous diseases [8,9]. Recently, ^1^O_2_ induced-DNA damage has been extensively discussed in the literature [10,11,12]. Researchers have based their work on the damage of DNA single/double-strands, and the four bases (adenine A, thymine T, cytosine C, and guanine G), caused by oxidation of in vivo and in vitro processes. Of the four DNA nucleobases, guanine has the lowest oxidation potential, and is the easiest to be oxidized by ROS [13]. The chemical structures of proposed guanine oxides are shown in Figure 1, of which 8-oxo-7, 8-dihydroguanine (8-oxoG) is widely known as a biomarker of oxidative stress [14,15]. It is considered that 8-oxoG, as an intermediate product, could undergo further oxidation by ^1^O_2_ to produce oxidized products such as spiroiminodihydantoin (Sp), guanidinohydantoin (Ghox), and 2,5- diamino-4H-imidazolone (Iz) [16,17,18].

Resveratrol (3,5,4-trihydroxy-trans-stilbene) is a natural phytoalexin found in grapes and wines, with health benefits including being antioxidant and anticarcinogenic. A mounting number of investigations have focused on the biological activities of its oligomers and analogues in recent years [19,20,21]. Previous studies revealed that Scirpusin A, a resveratrol dimer, is a selective scavenger of ^1^O_2_, and could protect DNA from ^1^O_2_ at low concentration, indicating that resveratrol dimers and its analogues could be potential candidates for the prevention of oxidative stress caused by photosensitization [22,23]. Our previous work suggests that resveratrol could relieve ^1^O_2_ induced oxidative damage and can conjugate guanine against ^1^O_2_ attack [24]. Nevertheless, the mechanism of resveratrol dimer protecting DNA against oxidative damage is still not clear. Since resveratrol dimer can effectively inhibit ^1^O_2_ induced DNA damage, it is important to investigate the mechanism in this content.

The primary objectives and aim of this work were to investigate the mechanism of resveratrol dimer preventing ^1^O_2_ induced DNA damage. We used the most susceptible base, guanine, to explore the prevention of DNA damage. Trans-ε-viniferin and trans-δ-viniferin were chosen as research subjects to investigate the inhibition of DNA damage. Qualitative and quantitative experiments were performed using UHPLC-QTOF-MS^2^ and UHPLC-QQQ-MS^2^ to monitor the changes in species and content of reactants and products during the process of resveratrol dimers’ protection of guanine against ^1^O_2_.

## 2. Materials and Methods

### 2.1. Reagents

Rose bengal (RB) was purchased from Ya’anda Biotechnology Co., Ltd. (Beijing, China). Tetrabutylammonium hydroxide (25%) was purchased from Aladdin (Shanghai, China). Trans-ε-viniferin (2.75 mmol/L dissolved in methanol) and trans-δ-viniferin (2.75 mmol/L dissolved in methanol) were prepared in our laboratory [25,26]. Chromatographic grade methanol used for UHPLC C-QTOF-MS^2^ qualitative analysis and UHPLC-QQQ-MS^2^ quantitative analysis was purchased from Merck (Darmstadt, Germany). Deionized water was purified with a MilliQ water system (Millipore, Bedford, MA, USA).

### 2.2. Sample Preparation

RB, as a photosensitizer, could absorb energy from ultraviolet light and transfer the energy to oxygen molecules to generate ^1^O_2_. The concentration of RB dissolved in deionized water during sample preparation was 72 µmol/L. Both the use and storage of RB were performed in a dimly lit environment. The concentration of stilbenes dissolved in methanol was 500 μmol/L. The guanine was dissolved in 1 mL 25% tetrabutylammonium hydroxide and 9 mL 5 mmol/L ammonium formate solution. Six groups of samples were prepared for experiments on UHPLC-QTOF-MS^2^ and UHPLC-QQQ-MS^2^, as previously described [27]. Six samples were freshly mixed at pH 4.5, and then the last three reaction mixtures were irradiated by a 365 nm UVA lamp for 5 min at 23 °C ± 2 °C. Before irradiation, the UVA light was turned on for at least 15 min to reach an equilibrium status. The first sample included 50 μL Guanine, 50 μL Rose Bengal, and 50 μL methanol (GRM); the second sample contained 50 μL Guanine, 50 μL Rose Bengal, and 50 μL trans-ε-viniferin (GRε-V); the third sample consisted of 50 μL Guanine, 50 μL Rose Bengal, and 50 μL trans-δ-viniferin (GRδ-V); the fourth UV irradiated sample contained 50 μL Guanine, 50 μL Rose Bengal, and 50 μL methanol (GRMU); the fifth UV irradiated sample consisted of 50 μL Guanine, 50 μL Rose Bengal, and 50 μL trans-ε-viniferin (GRε-VU); the last (UV irradiated) included 50 μL Guanine, 50 μL Rose Bengal, and 50 μL trans-δ-viniferin (GRδ-VU). We also measured two control experiments in the early stage, and no reactant was generated, and the guanine content did not change, so the data were not used.

### 2.3. Quantitative Analysis of Target Compounds by UHPLC-QQQ-MS^2^

The UHPLC-QQQ-MS^2^ system from Agilent Technologies consisted of a 1260-series UHPLC coupled to a triple quadrupole mass spectrometer. Samples were processed on a Thermo Scientific (Waltham, MA, USA) Acclaim^TM^ RSLC 120 C_18_ reversed-phase column (4.6 × 100 mm, 5 μm, 120 Å), using a gradient elution including ultra-pure water with 5 mmol/L ammonium formate added, and methanol with a flow rate of 0.4 mL/min at 20 °C. The injection volume was 2 μL. The multi-step gradient setting was as following: The initial condition was 5% B, held for 2 min; the percentage of B increased from 50% to 70% in the next 1 min, then, increased linearly from 70% to 90% over 5 min. Finally, the percentage of B went back to 5% in 2 min. The total running time was 10 min.

Mass spectrometric detection was performed on an Agilent 6460 QQQ instrument (Agilent, Santa Clara, CA, USA) equipped with an Agilent Jet Stream Technologies, Electrospray Ionization (AJS-ESI) source operating in negative ionization mode, with the conditions as follows: Nitrogen (99.99%) was used as the nebulizing and drying gas at 45 psi and a flow rate of 5.0 L/min, while the dry temperature was set as 350 °C. The temperature of sheath gas was 250 °C, and the flow was 11 L/min. The capillary voltage was 3500 V. The nozzle voltage was 500 V. Agilent Mass Hunter software was used to control the UHPLC-QQQ-MS^2^. The collision energy and fragmentor of target compounds had previously been optimized by molecular weights of precursor ions and product ions (Table 1) on Optimizer Software. A multiple reaction monitoring mode with the optimal fragmentor (V) and collision energy (eV) were used to quantify the content of the target compounds.

### 2.4. Qualitative Detection and Collection of Target Compounds by UHPLC-QTOF-MS^2^

The UHPLC-QTOF-MS^2^ experiments were carried out on a Thermo Scientific Dionex UltiMate 3000 system coupled with a Bruker micrOTOF-Q Ⅲ mass spectrometer (Bruker-Franzen Analytik GmbH, Bremen, Germany). The analysis of samples was performed on a Thermo Scientific AcclaimTM RSLC 120 C_18_ reversed-phase column (4.6 × 100 mm, 5 μm, 120 Å), with ammonium formate in ultra-pure water (5 mmol/L) and methanol as mobile phase, at a flow rate of 0.4 mL/min. The multi-step gradient elution conditions were set as following: The percent of B (*v/v*) increased linearly from 5% to 30% in the first 5 min, afterwards, it increased to 50% during next 5 min. Then, it increased to 90% during the next 5 min, and held for 5 min. The percentage of B went back to 5% in 7 min, and held for 5 min. The total running time was 32 min.

The induction of UHPLC effluent introduced into the ESI source by a solvent line (analytical, softron P/N 5040.8117). Software HyStar3.2 (Bruker Hyphenation Star Application, Germany) was used to control UHPLC and MS in combination. Experiments were performed under the negative ion mode of ESI. Nitrogen was used as the nebulizing and drying gas at 1.2 Bar and a flow rate of 8.0 L/min, and the drying temperature was set as 200 °C. Scan modes were of fragmentation amplitude and the configuration of collision energy at Auto MS/MS, with the mass scan range being 50–1000 *m/z*.

### 2.5. Data Analysis

Quantitative experiments were repeated three times and the data represented an average of these sets with standard error. One-way analysis of variance (ANOVA) followed by post-hoc test was performed to determine the statistical significance of the differences among different experimental groups using SPSS 18.0 software. A *p* value less than 0.05 was considered as statistically significant, and different letters indicated significant differences.

## 3. Results

### 3.1. Qualitative Analysis of Reactants and Products

In order to clarify the mechanism, UHPLC-QTOF-MS^2^ was used to identify the reactants and products in our investigations as the main research instrumental method. The base peak chromatogram of four samples are shown in Figure 2a,d and Figure 3a,d, respectively. As reported in Figure 2a,b, the molecular ion peak *m/z* 150.0435 corresponded to guanine, without UVA irradiated on the sample. Guanine preserved its structure and no oxidation occurred throughout the process (its secondary mass spectrometry is shown in Figure 2c). Quantitative ion was *m*/*z* 133 for guanine (*m*/*z* 150.0435 [M-H]^−^), which was used to quantify the peak intensity change of guanine in different reactions throughout this study.

As shown in Figure 2d,e, the precursor ion at *m/z* 182.0326 with high relative abundance was detected in GRMU. The 182.0326 [M-H]^−^ ion was identified as a representative ion of the oxidation product (its secondary mass spectrometry is shown in Figure 2f), and its quantitative ion determination was through *m*/*z* 139 (the secondary ion). In our study, the molecular ion peak *m*/*z* 150.0345, corresponding to guanine, was detected in all samples, and its retention time was 7.1 min. Sp (*m*/*z* 182.0326) was detected in the last three samples, which were treated with UVA irradiation, and the retention time of Sp was 2.9 min. The quantities of *m*/*z* 150 and *m*/*z* 182 were chosen as pivotal indicators to evaluate the degree of oxidative damage in guanine.

As shown in Figure 3, the ions of *m*/*z* 467.1134 (Figure 3a,b) and m/z 467.1118 (Figure 3d,e) were detected in GRε-VU and GRδ-VU, respectively. Yin et al. (2017) revealed the general mechanism of resveratrol dimers’ quenching ^1^O_2_ [27], where *m*/*z* 467 was considered as the product of the resveratrol dimer reacting with ^1^O_2_. This is consistent with our results (Figure 4), and the secondary mass spectrometry results are shown in Figure 3c,f.

Although *m*/*z* 467-1 and *m*/*z* 467-2 have similar product ions, they differ in the relative abundance of product ions, which means that they have different chemical structures. When the collision energy was 33.9 eV, the relative abundance of *m*/*z* 467.1134 was 87.8%, and the relative abundance of *m*/*z* 467.1118 was 100%. The difference in relative abundance of product ion was likely to be due to the difference in compound stability in collision-induced dissociation. Relatively, *m*/*z* 467-2 was more stable than *m*/*z* 467-1 in collision induced dissociation. The product ion at *m*/*z* 360 might be used to distinguish the two kinds of resveratrol dimer oxidations. The fragmentation pathway of the two resveratrol dimer oxidations is shown in Figure 5.

### 3.2. Quantitative Analysis of Reactants and Products (Characteristic Structure of Stilbenes Preventing ^1^O_2_ Induced DNA Damage)

To further investigate the mechanism of resveratrol dimers preventing ^1^O_2_-induced DNA damage, UHPLC-QQQ-MS^2^ was used to quantify the target compounds. After the parameters of fragmentor (V) and collision energy (eV) were optimized, the quantities of target compounds were evaluated by the abundance of their ion (peak intensity). The parameters of quantitative analysis are shown in Table 1.

As shown in Figure 6, compared with GRM, the quantity of guanine (*m*/*z* 150) in GRMU decreased significantly, while the quantity of Sp (*m*/*z* 182) in GRMU increased significantly, indicating that ^1^O_2_ can induce the oxidative damage of guanine. The quantities of guanine in GRε-VU and GRδ-VU were higher than that in GRMU, and *m*/*z* 467-1 and *m*/*z* 467-2 were found in GRε-VU and GRδ-VU, indicating that trans-ε-viniferin and trans-δ-vinferin quenched ^1^O_2_ selectively to inhibit ^1^O_2_ induced guanine damage.

## 4. Discussion

It is generally agreed that 8-oxoG is the main product of guanine oxidation [14,15]. However, since the 8-oxoG has a lower reduction potential than guanine, this makes it prone to further oxidative degradation to the compounds such as Sp [28]. Thapa et al. (2017) suggest that guanine could react with ^1^O_2_ to generate 8-oxoG, and then further oxidation could produce oxidized products such as Sp, Ghox, and Iz [16]. Our results point out that it is likely that 8-oxoG was oxidized further to Sp in a short time. This fact confirms that ^1^O_2_ can lead to guanine oxidative damage and the generation of Sp. Based on the results of qualitative analysis of reactants and products, guanine (*m*/*z* 150.0438), Sp (*m*/*z* 182.0343), oxide of trans-ε-viniferin (*m*/*z* 467.1134), and oxide of trans-δ-viniferin (*m*/*z* 467.1118) were selected as the target compounds for our quantitative study.

Quantitative results on guanine showed that trans-δ-vinferin is more effective than trans-ε-viniferin in impeding ^1^O_2_ induced guanine oxidation (Figure 6). We believe that the two kinds of resveratrol dimers could both prevent ^1^O_2_ induced DNA damage. In order to determine the characteristic structure which can prevent guanine oxidative damage, two kinds of resveratrol dimers (trans-ε-viniferin and trans-δ-vinferin) with the same concentration were selected in our investigation.

Our previous study showed that the phenol ring of resveratrol links with the free amino groups (-NH) of guanine at the beginning of ^1^O_2_ attack to form *m*/*z* 377.1104 [24]. However, the possible product of resveratrol dimers reacting with guanine has not been found in previous studies. Quantitative and qualitative analyses of this study showed that the content of *m*/*z* 182 was low in GRε-VU and GRδ-VU. In other words, ^1^O_2_ first reacts with resveratrol dimers but not guanine. Results showed that resveratrol dimers could protect guanine against ^1^O_2_. In the meantime, resveratrol dimers do not react with guanine, which would not affect the physiological activity of guanine.

As far as our results are concerned (Figure 4), the resorcinol ring is the characteristic structure of this reaction mechanism in the process of protecting DNA from oxidative damage, and it could be oxidized to quinones easily by ^1^O_2_ to prevent guanine oxidative damage. Furthermore, trans-δ-vinferin (which contains two resorcinol rings) inhibits the guanine oxidative damage more effectively than trans-ε-viniferin (which contains one resorcinol ring). Therefore, we hold the view that the resorcinol ring plays an important role in stilbenes protecting DNA against oxidative damage. Even though our experiments were performed under pH 4.5 and compounds went through mobile phase before being detected by the mass detector, the results still have biological relevance because there is little difference in the structural stability of guanine and resveratrol dimers between pH 4.5 and the physiological pH.

## 5. Conclusions

Resveratrol dimers can prevent ^1^O_2_ induced-DNA damage by replacing guanine to be oxidized by ^1^O_2_. Trans-δ-viniferin can inhibit DNA damage more effectively than trans-ε-viniferin. The mechanism is different from the resveratrol monomer, when guanine is attacked by ^1^O_2_. Resveratrol dimers do not bind with guanine to form a product, but their resorcinol reacts with the singlet oxygen to produce p-benzoquinone, thus protecting DNA from oxidative damage. This mechanism is significant for the development of health food, or medicine for the prevention and treatment of diseases related to DNA damage. Our results will facilitate the search for natural, non-toxic, and effective DNA protective agents, which could be used to combat diseases associated with DNA damage.

## Figures and Tables

**Figure 1 biomedicines-09-00271-f001:**
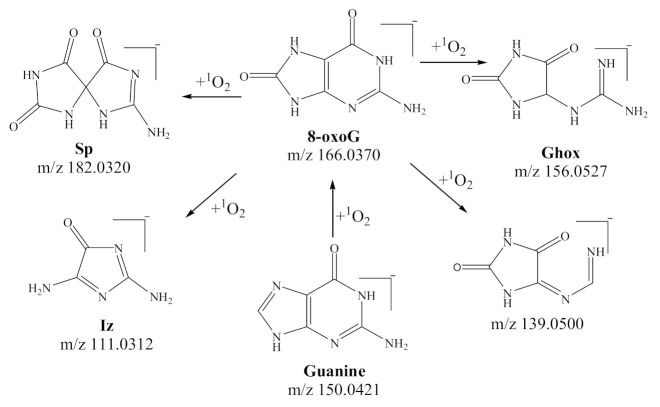
Proposed oxidative products of guanine induced by singlet oxygen in the literature (quoted from [16,24]).

**Figure 2 biomedicines-09-00271-f002:**
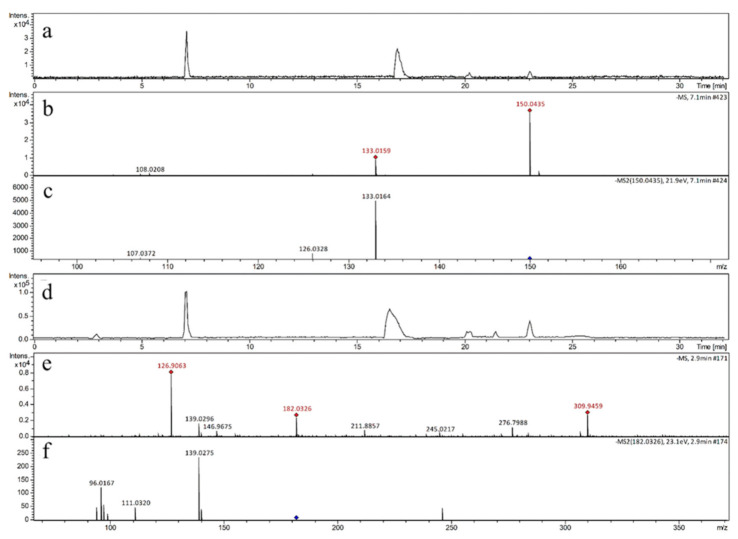
(**a**) The base peak chromatogram of the sample GRM; (**b**) The mass spectrum of GRM; (**c**)The MS/MS spectrum of *m*/*z* 150.0435; (**d**) The base peak chromatogram of the sample GRMU; (**e**) The mass spectrum of GRMU; (**f**) The MS/MS spectrum of *m*/*z* 182.0326.

**Figure 3 biomedicines-09-00271-f003:**
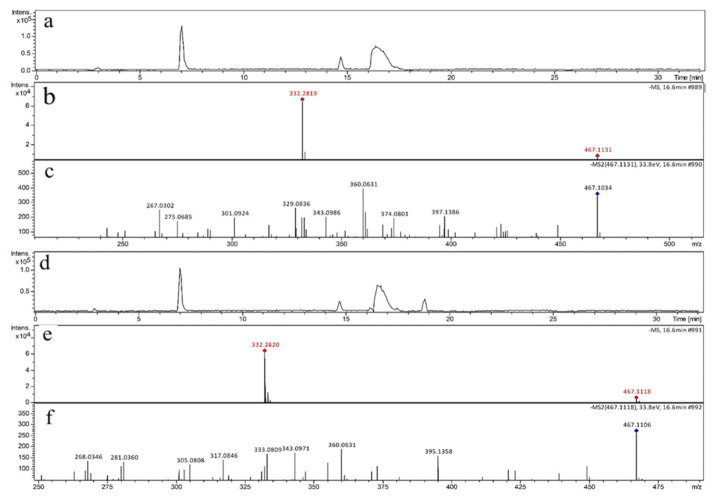
(**a**) The base peak chromatogram of the sample GRε-VU); (**b**) The mass spectrum of GRε-VU; (**c**) The MS/MS spectrum of *m*/*z* 467.1134 (*m*/*z* 467-1); (**d**) The base peak chromatogram of the sample GRδ-VU; (**e**) The mass spectrum of GRδ-VU; (**f**) The MS/MS spectrum of *m*/*z* 467.1118 (*m*/*z* 467-2).

**Figure 4 biomedicines-09-00271-f004:**
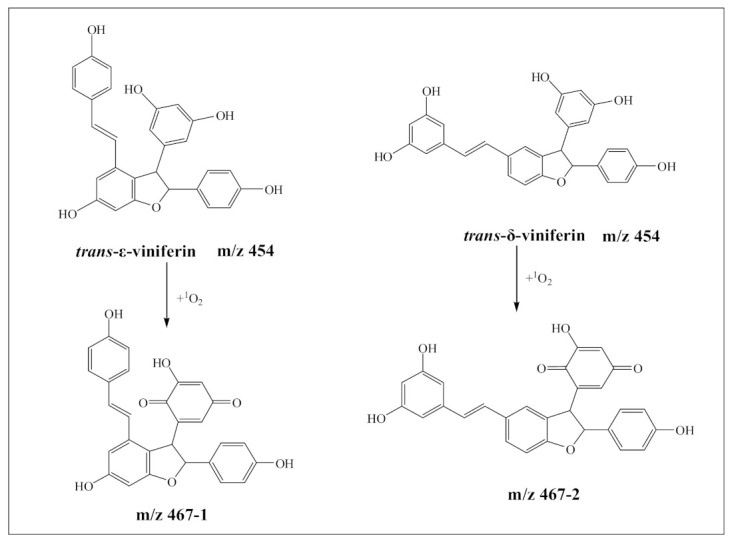
Proposed oxidative products of resveratrol dimers induced by singlet oxygen.

**Figure 5 biomedicines-09-00271-f005:**
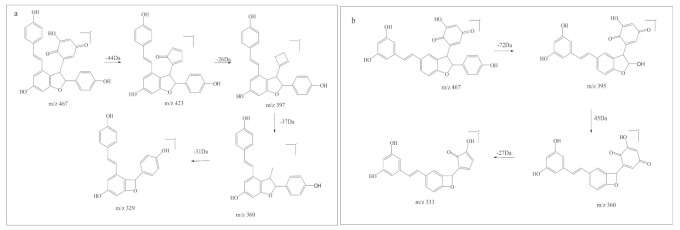
The fragmentation pathway of the two resveratrol dimer oxidations. (**a**) trans-ε-viniferin + ^1^O_2_; (**b**) trans-δ-viniferin + ^1^O_2_.

**Figure 6 biomedicines-09-00271-f006:**
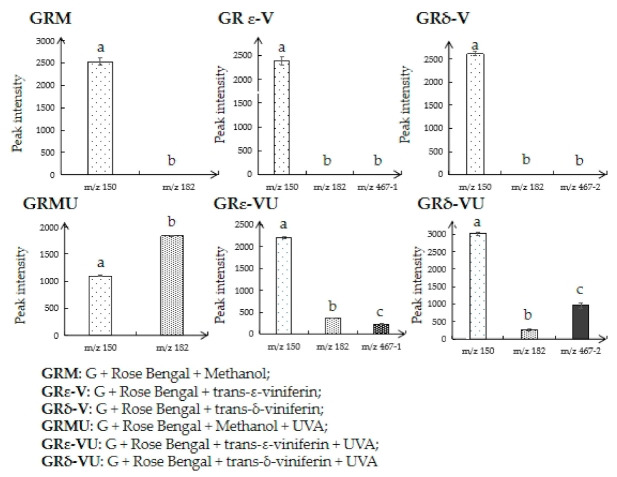
Quantitative results of target compounds in different treated groups. (Different letters indicate significant differences, i.e., a is significantly different to b or c, while data with the same letter, i.e., b vs. b, do not have significant difference).

**Table 1 biomedicines-09-00271-t001:** The optimal parameters of target compounds for Multiple Reaction Monitoring (MRM) mode.

Precursor Ion	Product Ion	Fragmentor (V)	Collision Energy (eV)
*m/z* 150	*m/z* 133	60	10
*m/z* 182	*m/z* 139	60	6
*m/z* 467-1	*m/z* 360	165	20
*m/z* 467-2	*m/z* 360	165	25

467-1. Possible product of trans-ε-viniferin reacting with singlet oxygen; 467-2. Possible product of trans-δ-viniferin reacting with singlet oxygen.

## Data Availability

The data presented in this study are available on request from the corresponding author. The data are not publicly available due to possible commercial applications.

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
