# Peer review of "Mechanism of Resveratrol Dimers Isolated from Grape Inhibiting 1O2 Induced DNA Damage by UHPLC-QTOF-MS2 and UHPLC-QQQ-MS2 Analyses"

_biomedicines, 2021, doi:10.3390/biomedicines9030271_

Round 1
Reviewer 1 Report
The authors have addressed my second and thirds comments but not my first comment which was.
- I wrote "The experimental design lacks two control experiments which need to be carried out ie guanine + trans-e-viniferin (ie no Rose Bengal) and guanine + trans-d-viniferin (again no Rose Bengal)." Their response is fine but they should modify and add to the text their following statement. "We also measured two control experiments in the early stage, and no reactant was generated, and the guanine content did not change, so the data was not used."
The author's response as "The quantitative results of the two control experiments have been added to the manuscript and Figure 6." Unfortunately I can not see any such material in the manuscript or in Figure 6 (as the data in Figure 6 only shows treatments with rose bengal)
Author Response
We appreciate very much the reviewer's comment. We have added the suggested sentence "We also measured two control experiments in the early stage, and no reactant was generated, and the guanine content did not change, so the data was not used." in the manuscript, and highlighted with yellow.
Reviewer 2 Report
The manuscript is notably improved. Figure 6 is more convincing. I have found a typo at page 109: please, change epsilon to delta.
In the caption to Fig.6., please explain what differences the letters a, b and c are meaning. In other words, what values are compared, b with b, and b with c? It is not clear.
Author Response
We'd like to thank reviewer for the comments. Very much appreciated!
We have fixed the typo (now at line 103 highlighted in yellow)
We have added explanation of the meaning of the letters in the figure legend and highlighted in yellow (Different letters indicated significant differences, i.e. a is significantly different to b or c, while data with same letter, i.e. b vs. b, do not have significant difference).
This manuscript is a resubmission of an earlier submission. The following is a list of the peer review reports and author responses from that submission.
Round 1
Reviewer 1 Report
Review of the manuscript:
Mechanism of isomers and analogues of resveratrol dimers selectively quenching singlet oxygen by UHPLC-ESI-MS2
Xuefeng Yin, Jia Yu, Qingjun Kong, Xueyan Ren
The studies were focused on the role of resveratrol dimers: trans-ε-viniferin and trans- δ-viniferin in the protection of guanine against oxidation by singlet oxygen. The structures of oxidized dimers of resveratrol formed in the reaction with singlet oxygen generated by photosensitized Rose Bengal (4,5,6,7-tetrachloro-2',4',5',7'-tetraiodofluorescein) were identified with the use of UHPLC-QTOF-MS method. The authors refer to the results earlier reported (Yin et al., 2017). In the present study, the role of resveratrol dimers as 1O2 quenchers was investigated in the chemical system containing guanine, tested resveratrol dimers and Rose Bengal + UVA as a source of singlet oxygen. The results indicate (Figure 6), that in the presence of resveratrol dimers guanine is protected against oxidation caused by singlet oxygen.
- Definitely, this manuscript should be improved in terms of comprehensibility. Too often I had to guess what authors wanted to say. Even the title is formulated in ambiguous way. In my understanding, “via UHPLC” suggest the analytical method was involved in the guanine oxidation. The language of the manuscript makes very difficult to understand the contents and lowers the overall quality of the manuscript.
I give only some examples of incomprehensive sentences and recommend to check the whole text with regard to comprehensibility and English phrasing.
To the sentence in lines 21-22: guanine and dimers were attacked with singlet oxygen, and mostly dimers were oxidized, in this way protecting guanine against oxidation.
Lines 23-25: what is different from resveratrol monomer?
Lines 31-32: authors should find a better definition of oxidative stress.
Line 149: What does it mean “indicated by dissimilar alphabets a, b, c, d “?
- The most important Figure 6 in my opinion would be more convincing if the histograms A, B, C and D demonstrate four studied chemical systems:
1) Guanine + water + methanol, and in my opinion, this mixture should be also irradiated with UVA;
2)G + Rose Bengal + Methanol + UVA;
3) G + Rose Bengal + trans-ε-viniferin + UVA;
4) G + Rose Bengal + trans- δ-viniferin + UVA.
Differences statistically significant are badly presented.
- Relative abundance shown in Fig. 6 should be defined because it is not clear why at the Y axis are so high values (>100%).
- There are many mistakes in References
Author Response
Replies to reviewer
- The language of the manuscript have made careful modifications.
Especially, the title of this manuscript has been revised, “via” has been modified to “by”.
The sentence “guanine and dimers were attacked with singlet oxygen, and mostly dimers were oxidized, in this way protecting guanine against oxidation” has been added to Lines 21-22.
Lines 23-25: The sentence“what is different from resveratrol monomer” have been deleted.
Lines 31-32: The definition of oxidative stress has been revised to “Oxidative stress refers to that the oxidative and anti-oxidative effects in the bodys are out of balance, and tends to be oxidized, which is a negative effect produced by free radicals”.
Line 149: The expression of “indicated by dissimilar alphabets a, b, c, d” has been modified to “different letters indicated significant differences”.
- The experimental group in Figure 6 has been modified and reprocessed.
- The value of the Y axis in the original figure is the peak area of the compound measured by UHPLC-QQQ-MS2, which has beenconverted into relative content.
- Referenceshave been revised in according to 'Instructions for Authors'.
Reviewer 2 Report
This manuscript describes the MS identification of products arising from the quenching of 1O2 induced guanine oxidation products by the resveratrol dimers (trans-e-viniferin and trans-d-viniferin) and suggests that 1O2 reacts with he dimers to produce p-benzoquinone.
There are some major issues that need to be resolved in both the description of the sample preparation and experimental design and presentation of the quantitative analysis.
For the sample preparation:
- What is the concentration of guanine and the relative concentrations of guanine: viniferins as this will no doubt determine the levels of trapping of 1O2 by the viniferins and reduction in guanine oxidation.
- The experimental design lacks two control experiments which need to be carried out ie guanine + trans-e-viniferin (ie no Rose Bengal) and guanine + trans-d-viniferin (again no Rose Bengal).
- It is unclear what is the pH of the reaction mix - if not physiological then the experiments need to be repeated at a physiological pH
There are some issues in the presentation of the quantitative data. The applicants determine the relative content of the target compounds (line 121). However, Figure 6 present the results as if they were absolute amounts by encouraging a comparison across the different experiments in each panel which is an incorrect comparison since the values are relative within an experimental group (and not across experimental groups). Panels must present results from individual experimental groups.
In addition, the authors need to come up with better terminology than just first/second/third/fourth sample to make it clear within the text what these samples are without the necessity for returning to the definitions each time.
Additional issues to be clarified:
Line 30. “Oxidative stress is an in-house defense mechanism of antioxidant systems” is an incorrect definition of oxidative stress.
Line 41. “DNA is considered to be a collection..” again is incorrect “DNA is a collection….”
Figure 1. “Ghox” is not defined; name of the chemical in the bottom right of the Figure is not given.
Lines 193/194. “And the relative abundance of m/z 360.0631-1 (100%, product ion of m/z 467-1) was higher …”. Where is this data? I can’t see any figures where the compound with an m/z 360.0631-1 is the total product (ie 100%)
Author Response
Replies to reviewer
For the sample preparation:
- The concentration of guanineis 5 mmol/L and the relative concentrations of guanine: viniferins is 10: 1.
- This experiment focuses on the damage to guanine in the presence of 1O2and the protective effect of trans-e-viniferin and trans-d-viniferin on guanine, mainly the tracking and monitoring of guanine and three reaction products to derive the reaction mechanism. We also measured two control experiments in the early stage, and no reactant was generated, and the guanine content did not change, so the data was not used.
- This experiment temporarily discusses the in vitro mechanism of resveratrol dimer inhibiting 1O2-induced DNA damage, and does not involve the factor of pH. The subsequent deeper research will be carried out at physiological pH.
For the presentation of the quantitative data:
The value of the Y axis in the original figure 6 is the peak area of the compound measured by UHPLC-QQQ-MS2, has been processed according to the relative content of the target compound.
The samples have also been replaced with new terminology.
For the additional issues:
Line 30 the definition of oxidative stress have been revised to “Oxidative stress refers to that the oxidative and anti-oxidative effects in the bodys are out of balance, and tends to be oxidized, which is a negative effect produced by free radicals”
Line 41 “DNA is considered to be a collection..” have been revised to “DNA is a collection….”.
Figure 1. “Ghox” was defined and the name of the chemical in the bottom right of the Figure have been given.
Lines 193/194 refers to Figure 3c, m/z 360.0631 is the key product ion in the MS/MS spectrum of m/z 467-1 with the highest content, and its relative abundance defaults to 100%, the relative abundance values of other product ions are relative to this. In Figure 3f, the content of 467.1106 in the MS/MS spectrum of m/z 467-2 is the highest, and its relative abundance is 100%, compared with this, the relative abundance of m/z 360.0631 is 72.3%.

Round 2
Reviewer 1 Report
The manuscript is still roughly written, some examples:
Line 167: What does it mean: “without UV-irradiated in sample”
Line 248: I guess that the authors mean “resveratrol dimers”
Line 250: …study shows….
The Figure 6 is still illegible. What does the relative abundance/content mean is not explain in the caption and in the section Methods. In the text relative abundance is expressed in % in the Figure 6 in thousands of relative units.
Figure 6 C and D: What is the sense to determine statistically the significance of difference in the content of oxidized viniferins with the sample not containing viniferin at all (was not added from the beginning)?
As I advised the authors earlier, in the distinct histograms the identified species found in four groups: GR-M, GR-MU, GR-ε-VU, GR-δ-VU should be demonstrated: Fig. 6 A – GR-M; Fig. 6 B – GR-MU; Fig. 6 C – GR-ε-VU; Fig. 6. GR-δ-VU
Line 104-105: explain the difference between GR-M and GR-MU. For a reader it is exactly the same! In the text is written that all four samples were irradiated (line 107-110).
Reviewer 2 Report
The authors have addressed most but not all of my comments.
- I wrote "The experimental design lacks two control experiments which need to be carried out ie guanine + trans-e-viniferin (ie no Rose Bengal) and guanine + trans-d-viniferin (again no Rose Bengal)." Their response is fine but they should modify and add to the text their following statement. "We also measured two control experiments in the early stage, and no reactant was generated, and the guanine content did not change, so the data was not used."
- the authors did not address the comment "It is unclear what is the pH of the reaction mix - if not physiological then the experiments need to be repeated at a physiological pH " save to say that "The subsequent deeper research will be carried out at physiological pH." So at present we don't know whether the mechanism proposed here is of any relevance.
- the authors have misunderstood my comment about Figure 6. The change of the y axis from "relative content" to "relative abundance" does not address the issue. Since the authors themselves indicate that the abundance is determined within an experiment, the lack of any internal controls means that the relative abundance across different experiments cannot be compared. Ie Figure 6 can have four panels but each panel must contain the relative abundance of the different compounds from a single experiment and not the relative abundance of the same compound from different experiments.